# Effects of Seven Plant Essential Oils on the Growth, Development and Feeding Behavior of the Wingless *Aphis gossypii* Glover

**DOI:** 10.3390/plants13070916

**Published:** 2024-03-22

**Authors:** Xinhang Wang, Ying Zhang, Haibin Yuan, Yanhui Lu

**Affiliations:** 1Department of Plant Protection, Jilin Agricultural University, Changchun 130118, China; wangxinhang0302@163.com; 2State Key Laboratory for Biology of Plant Diseases and Insect Pests, Institute of Plant Protection, Chinese Academy of Agricultural Sciences, Beijing 100193, China; yingzhang6565@163.com; 3Western Agricultural Research Center, Chinese Academy of Agricultural Sciences, Changji 831100, China

**Keywords:** essential oil, *Aphis gossypii* Glover, antifeedant activity, growth and development

## Abstract

Cotton aphid *Aphis gossypii* Glover damages plants such as cotton directly by feeding on leaves and indirectly by transmitting viruses and excreting honeydew, which interferes with photosynthesis. The control of *A. gossypii* is still dominated by the frequent use of insecticides, which leads to a gradual increase in pesticide resistance in *A. gossypii*. Research is therefore needed on non-pesticide controls. In this study, seven plant essential oils (EOs) of *Ocimum sanctum* L., *Ocimum basilicum* L., *Ocimum gratissimum* L., *Mentha piperita* L., *Mentha arvensis* L., *Tagetes erecta* L., and *Lavandula angustifolia* Mill. were examined as potential controls for *A. gossypii*. We used life tables and electrical penetration graphs (EPG) to explore the effects of these EOs on the growth, development, and feeding behavior of *A. gossypii*, followed by a study of effects of the EOs on honeydew secretion by *A. gossypii* as a measure of their antifeedant activity. We found that the EOs of *O. sanctum*, *M. piperita*, *M. arvensis* and *T. erecta* significantly extended the pre-adult developmental period. Also, adult longevity, number of oviposition days, and total fecundity of *A. gossypii* treated with the EOs of *M. arvensis* or *T. erecta* were all significantly reduced. *Aphids* treated with the EOs of *O. sanctum*, *M. piperita*, or *L. angustifolia* showed significant reductions in their net reproductive rate (*R*_0_), intrinsic rate of increase (*r*_m_), and finite rate of increase (*λ*), and significant increases in mean generation time (*T*). In terms of their effects on the feeding behavior of *A. gossypii*, all seven EOs significantly reduced the total duration of phloem feeding (E2 waves), the number of phloem-feeding bouts, and the proportion of time spent in secretion of saliva into phloem sieve elements (E1 waves) and phloem feeding (E2). The total duration and number of E1 waves (saliva secretion) were significantly reduced by the EOs of *O. sanctum*, *O. gratissimum*, and *M. arvensis*. For C waves (probing in non-vascular tissues), the total duration spent in this behavior was significantly increased by the EOs of *O. gratissimum*, *M. piperita*, and *L. angustifolia*, but the number of such probing events was increased only by *L. angustifolia* EO. The EOs of *O. basilicum*, *M. arvensis*, and *T. erecta* significantly increased the total duration of ingestion of xylem sap (G waves), while the total time of mechanical difficulty in stylet penetration (F waves) was increased by *M. arvensis*. The total duration and number of the non-probing events (Np waves) were significantly increased by EOs of *O. sanctum* and *O. basilicum*. After treatment with all seven of these EOs, the area covered by honeydew was significantly reduced compared with the control. Studies have analyzed that EOs of *O. sanctum*, *M. piperita,* and *T. erecta* were most effective, followed by the EOs of *M. arvensis* and *L. angustifolia*, and finally the EOs of *O. basilicum* and *O. gratissimum*. In the present study, the EOs of *O. sanctum*, *M. piperita,* and *T. erecta* were found to have potential for the development as antifeedants of *A. gossypii*, and these data provide a basis for future research on non-pesticide chemical control of *A. gossypii*.

## 1. Introduction

Cotton aphid *Aphis gossypii* Glover (Hemiptera: Aphididae) has a wide range of hosts and is harmful to cotton, potato, and melons, among others [1]. Its damage includes direct feeding injury to foliage and stems and mission transmission of plant viruses [2], including cotton leafroll dwarf virus [3,4]. Wingless aphids develop faster than winged aphids. Wingless aphids have adaptations that maximize fecundity, making them more difficult to control [5]. Insecticides are still the main method for control of *A. gossypii* in most agricultural settings. However, past frequent use of insecticides has resulted in *A. gossypii* populations resistant to all the major insecticide groups [6]. Nauen et al. showed that field populations of *A. gossypii* and *Myzus persicae* Sulzer in several European countries were highly resistant to pirimicarb and oxydemeton-methyl [7]. Patima et al. demonstrated that *A. gossypii* populations in cotton fields in several regions of Xinjiang, China, have developed very high levels of resistance to deltamethrin, cypermethrin, and omethoate [8]. Insecticide use may also harm populations of important natural enemies of pest insect [9], contributing in this case to resurgence of aphid populations. Therefore, research and development of non-chemical pesticides for cotton aphid control are needed.

Plant essential oils (EOs) are volatile, naturally occurring, complex compounds found in aromatic plants, where they act as secondary metabolites. EOs intended for commercial uses can be collected by distillation, mechanical pressure, or extraction [10]. EOs have potential as alternatives to commercial synthetic pesticides for use in green agriculture. Karamanoli demonstrated that EOs of spearmint and peppermint in soil declined about 90% within 30 days [11]. Because of such rapid degradation of EOs in nature, they do not accumulate in the environment and thus leave no polluting residues. Some botanical insecticides such as pyrethrum and neem are already well established as commercial pest control products [12], and are recommended for control of insects such as aphids, mirid bugs and whiteflies on field crops, as well as pests of glasshouse crops, house plants, and stored products [13]. Today, more than 100 commercial neem formulations are used worldwide [14]. Insecticides from the oils of citrus peel, containing limonene, have recently had moderate commercial success in North America and Europe [15]. Following neem oil, orange oil was the most frequently used botanical insecticide in California in 2016, with 7.0 tons of active ingredient being applied [16]. Additionally, the EOs of *Chenopodium ambrosioides* L. are used for insect pest management [17], with an average of 8000 kg sold per year in California, United States [15].

Plant EOs have great potential in the control of aphids. A number of articles have been published both at home and abroad reporting the fumigant and contact activities of EOs in the Lamiaceae and Asteraceae families against aphids. The EOs of *Mentha piperita* L., *Mentha pulegium* L., and *Ocimum basilicum* L. showed excellent efficacy in both contact and fumigation tests against *Lipaphis pseudobrassicae* Davis, *M. persicae*, and *A. gossypii* [18]. *Lavandula angustifolia* Bubani oils had fumigant activity against *Acyrthosiphon pisum* Harris [19]. *Mentha longifolia* L. oils had fumigant and contact activities against *Aphis craccivora* Koch, and affected the development, survival, and reproduction of *A. craccivora* [20]. *Artemisia dracunculus* L. had high fumigant toxicity to *A. gossypii* [21]. The EOs *Santolina chamaecyparissus* L. and *Tagetes patula* L. showed contact activity against *Myzus persicae* Sulzer and *Rhopalosiphum padi* L., while sublethal concentrations of the two essential oils reduced aphid fecundity [22].

At present, the EOs of Lamiaceae and Asteraceae have been reported to have antifeedant activity against a variety of aphids. Abualfia et al. discovered that the EOs of *Salvia officinalis* L., *Lavandula spic*a L. and *Mentha spica* L. caused different degrees of interference with aphid feeding [23]. Zhang et al. found that the ethanol extract of *Xanthium strumarium* L. had stronger antifeedant activity against aphids feeding on *Lycium barbarum* L., with a plant rejection rate of more than 82% [24]. Zhou demonstrated that *Flaveria bidentis* L. oil had different levels of antifeedant activity on *Lipaphis erysimi* Kaltenbach and *Rhopalosiphum maidis* Fitch [25]. However, research on the antifeedant activity of plant EOs on aphids is relatively scarce. We sought to determine whether the EOs from Lamiaceae and Asteraceae plants exhibit antifeedant effects on cotton aphids, which will provide a theoretical basis for the development of subsequent antifeedant.

In this study, we examined the effects of EOs from seven plants (*Ocimum sanctum* L., *Ocimum basilicum* L., *Ocimum gratissimum* L., *Mentha piperita* L., *Mentha arvensis* L., *Tagetes erecta* L., and *Lavandula angustifolia* Mill.) on the growth, development and fecundity of *A. gossypii*. In addition, we analyzed the feeding behaviors of *A. gossypii* using the EPG technique. Lastly, we measured honeydew excretion by aphids feeding on cotton leaves treated with EOs of seven plant species, using honeydew excretion as a measure of aphid feeding.

## 2. Results

### 2.1. Effect of EOs on Cotton Aphid Growth, Development, and Fecundity

The developmental durations of all nymphal instars of *A. gossypii*, except the third instar, were affected by exposure to plant EOs (Table 1). Duration of the first instar was significantly prolonged by three plants: *O. basilicum* (*p* = 0.002), *M. arvensis* (*p* = 0.005), and *T. erecta* (*p* = 0.004). Similarly, the stage duration of second instars was affected by three different plant EOs: *O. sanctum* (*p* = 0.004), *M. piperita* (*p* < 0.001), and *L. angustifolia* (*p* = 0.012). For fourth instars, stage duration was affected by two plants—*O. basilicum* (*p* = 0.002) and *O. gratissimum* (*p* = 0.024)—while for all instars as a group, four plants’ EOs had significant effects: *O. sanctum* (*p* = 0.003), *M. piperita* (*p* = 0.001), *M. arvensis* (*p* = 0.003), and *T. erecta* (*p* = 0.042).

The longevity and fecundity of *A. gossypii* treated with different plant EOs were generally shortened by plant EOs (Table 2). The EOs of all seven plants significantly reduced adult longevity (*p* < 0.001). Aphid fecundity (as number of offspring produced) was reduced by the EOs of five plants: *O. sanctum* (*p* < 0.001), *M. piperita* (*p* < 0.001), *M. arvensis* (*p* = 0.011), *T. erecta* (*p* = 0.001), and *L. angustifolia* (*p* < 0.001). The number of reproductive days for adult aphids was shortened by the EOs of four plants: *O. basilicum* (*p* = 0.001), *O. gratissimum* (*p* < 0.001), *M. arvensis* (*p* < 0.001), and *T. erecta* (*p* < 0.001).

Compared with the control group, *A. gossypii* treated with EOs of three plants (*O. sanctum*, *M. piperita,* and *L. angustifolia*) showed significant reductions in net reproductive rate (*R*_0_), intrinsic rate of increase (*r*_m_), and finite rate of increase (*λ*), and significant increases in mean generation time (*T*) (Table 3).

### 2.2. Effect of Plant EOs on Aphid Feeding Behavior

Plant EOs showed effects on various non-phloem feeding behaviors in *A. gossypii* (Table 4). Except for the total duration of probing inside cells (Pd waves), which should not be affected, the seven EOs not only lengthened the total duration of probing in non-vascular tissues (C waves), ingestion of xylem sap (G waves), and non-probing activity (Np waves). EO exposure also increased the number of probing events in non-vascular tissues (C waves) and of non-probing events (Np waves). For the behavior of probing in non-vascular tissues (C waves), the total duration was significantly increased by *O. gratissimum* (*p* = 0.019), *M. piperita* (*p* = 0.013), and *L. angustifolia* (*p* = 0.004). For *L. angustifolia*, the number of such events, but not their total duration, was increased (*p* = 0.002). The EOs of *O. basilicum* (*p* = 0.021), *M. arvensis* (*p* = 0.001), and *T. erecta* (*p* < 0.001) significantly increased the total duration of xylem feeding (G waves). The EO of *M. arvensis* also increased the total duration of mechanical difficulty in stylet penetration (F waves) (*p* = 0.011). The total duration of the non-probing behavior (Np waves) was significantly increased by *O. sanctum* (*p* = 0.002), *O. basilicum* (*p* = 0.002), and *O. gratissimum* (*p* = 0.025). The number of non-probing behavior events (Np waves) was significantly increased by *O. sanctum* (*p* = 0.001) and *O. basilicum* (*p* = 0.010).

Plant EOs also, importantly, affected the phloem-feeding behaviors of *A. gossypii* (Table 5). All seven EOs significantly reduced the total duration of phloem ingestion (E2 waves) (*p* = 0.001), the number of phloem-feeding events (*p* = 0.045), and the proportion of time occupied by saliva secretion and phloem feeding combined (E1 and E2 waves) (*p* < 0.001). However, for the other behaviors, each essential oil had different effects. The total duration of salivation (E1 waves) was significantly reduced by *O. sanctum* (*p* = 0.002), *O. basilicum* (*p* = 0.025), *O. gratissimum* (*p* = 0.017), *M. arvensis* (*p* = 0.001), and *L. angustifolia* (*p* = 0.004). Also, the number of salivation events (E1 events) was significantly reduced by *O. sanctum* (*p* = 0.025), *O. gratissimum* (*p* = 0.029), and *M. arvensis* (*p* = 0.004). The EOs of *O. sanctum* (*p* = 0.026) and *T. erecta* (*p* = 0.022) also significantly prolonged the time to the first salivation event (E1), but had no effect on the time to the first phloem feeding (E2). 

### 2.3. Effect of Plant EOs on Daily Secretion of Aphid Honeydew

In the experiment, the honeydew distribution was scattered and no large spots were formed, so the error caused by stacking could not be considered. Exposure to plant EOs significantly reduced the level of honeydew excretion by *A. gossypii* compared with the control (*p* < 0.001) (Table 6). The levels of antifeedant impact (based on the reduction in honeydew by a treated aphid relative to a control aphid) due to EOs of *O. basilicum*, *O. gratissimum*, *M. piperita*, *M. arvensis*, and *T. erecta* were 64.6%, 51.1%, 50.7%, 65.2%, and 59.2%.

## 3. Discussion

We examined the effects of seven plant EOs (*O. sanctum*, *O. basilicum*, *O. gratissimum*, *M. piperita*, *M. arvensis*, *T. erecta*, and *L. angustifolia*) on the growth, development, feeding behavior, and honeydew excretion of *A. gossypii*. All seven EOs had varying effects on *A. gossypii*, with the most effective from the point of view of aphid control being the EOs of *O. sanctum*, *M. piperita*, and *T. erecta*.

In general, it is the case that plant EOs can adversely affect the growth and development of aphids. For example, the EO of *Tagetes minuta* L. significantly reduced the fecundity of *Acyrthosiphon pisum* Harris, *M. persica*e, and *Aulacorthum solani* Kaltenbach [26]. The EOs of *Origanum majorana* L., *Mentha pulegium* L., and *Melissa officinalis* L. significantly reduced the longevity and fecundity of *M. persicae* [27]. We showed that the adult longevity of *A. gossypii* was significantly shortened and its fecundity significantly reduced by the EOs of six of our seven test plants (*O. sanctum*, *O. gratissimum*, *M. piperita*, *M. arvensis*, *T. erecta*, and *L. angustifolia*). Also, the net reproductive rate (*R*_0_), intrinsic rate of increase (*r*_m_), and finite rate of increase (*λ*) of *A. gossypii* were all significantly reduced, and the mean generation time (*T*) was significantly increased after treatment with EOs of *O. sanctum*, *M. piperita* and *L. angustifolia*. The growth, development, and fecundity of *A. gossypii* were all diminished, possibly due to a reduction in time spent in phloem and xylem feeding, reducing the level of nutrition available for growth [28].

Insects assess whether a plant is suitable for feeding by sampling plant parts for their nutrients [29]. Aphid probing and prolonged feeding are closely associated with the level of plant tissue damage and plant virus transmission [30]. *A. gossypii* obtains the nutrients it needs for growth and development mainly by feeding on the phloem sap [31]. Aphids promote the success of such feeding by secreting water-soluble saliva, which can prevent plant phloem proteins from clogging sieve tubes [32]. In our experiment, we observed that secretion of saliva (E1) and phloem feeding (E2 waves) by *A. gossypii* were both reduced after exposure to EOs, leading to less successful feeding by *A. gossypii*. Similarly, for pea aphids (*A. pisum*) treating its host plants with kaempferol led to an increase in the time spent in non-probing behaviors [33]. When plants infested with *A. gossypii* were treated with imidacloprid, *A. gossypii* spent significantly longer time on intercellular apoplastic stylet pathway activities, which are non-production behaviors taking time away from feeding [34]. For *Diaphorina citri* Kuwayama non-probing behaviors (Np) were prolonged after exposure to the EOs of guava (*Psidium guajava* L.) [35]. Similarly, in our study, *A. gossypii* spent more time in non-probing behaviors after exposure to the plant EOs examined here. Also, the duration and number of events of non-vascular probing (C waves) increased in *A. gossypii* following exposure to our test EOs, indicating that more probing occurred between the plant epidermis and the microtubule bundles, requiring more time to find a suitable feeding site. Zhou et al. observed plants’ aqueous extracts of *Nerium indicum* Mill., *Cinnamomum camphora* L., *Ginkgo biloba* L., and *Lycopersicon esculentum* Mill. significantly prolonged the duration of non-vascular probing in *A. gossypii* [36]. In contrast to phloem ingestion, xylem ingestion supplies water but not nutrition, and increased time spent in xylem feeding may be linked to a decrease in phloem feeding [37]. Aphids spend more time xylem feeding when under stress [38]. We found the same behavior in our experiment, in which the xylem feeding was prolonged after treatment with EOs, presumably to maintain the internal water equilibrium [39].

After aphids feed on phloem, excess water, which still contains some sugar, is excreted as honeydew to concentrate sugars in the fluids retained for ingestion [40]. In contrast, xylem feeding results in little or no honeydew production because most ingested xylem is retained to maintain water equilibrium [41]. Honeydew excretion is, therefore, closely tied to the level of phloem feeding by an aphid [42]. Honeydew often covers plant surfaces and affects photosynthesis and respiration [43]. When an aphid’s level of phloem feeding is reduced, its rate of growth and development declines, and it excretes less honeydew. By quantifying the volume of honeydew excreted by cotton aphids, we were able to determine the degree to which plant essential oils had influenced the feeding of cotton aphids. Extracts of *Syngonium podophyllum* Schott, *Xanthium sibiricum* Petr.et Widd, and *Tephrosia vogelli* Hook have been shown to reduce honeydew excretion by *M. persicae* and *Lipaphis erysimi* Kaltenbach [44]. In our study, the daily rate of honeydew excretion by *A. gossypii* declined significantly after treatment with plant EOs. This decline in honeydew production indicates that plant EOs have significant antifeedant effects on *A. gossypii*, but the EOs of different plants varied in the levels of their antifeedant effects on *A. gossypii*. We observed that the honeydew of *A. gossypii* was dispersed on the filter paper, likely indicating that the EO-treated leaves were not acceptable for aphid feeding, and aphids kept moving and searching for better sites. This suggests that the honeydew production of *A. gossypii* was reduced and no suitable sites were located, given that there was no overlapping of honeydew to from blotches on the filter paper.

From an integrated pest management point of view, when considering the effects of EOs on pest control, we also need to examine their effects on natural enemies. Chemical control not only leads to an increase in pest resistance but also kills large numbers of natural enemies, weakening their contribution to pest control, which may lead to pest outbreaks. It has, however, been demonstrated that many EOs are much less damaging to natural enemies than to the target pests. Papadimitriou et al. compared the biological activity of the EO of *M. pulegium* against *A. gossypii*, *Tetranychus urticae* Koch, and the mirid *Nesidiocoris tenuis* Reuter, and found that the plant’s EO was highly lethal to both *A. gossypii* and *T. urticae* at a concentration of 1000 μL/L, but had no toxic effects on *N. tenuis* [45]. In another study, the LC_50_ values of the EOs of *M. piperita*, *Mentha longifolia* L., *Salvia officinalis* L., and *Salvia rosmarinus* Spenn. were about four times higher against the coccinellid *Coccinella undecimpunctata* L. than to the pest, *Aphis punicae* Passerini [46]. Ebadollahi et al. observed that the EO of *Satureja intermedia* C.A.Mey showed significant contact activity against *Aphis nerii* Kaltenbach, but was safe against its predator *Coccinella septempunctata* L. [47]. These findings suggest that insect natural enemies may be more tolerant to a wide range of plant EOs than are target pests.

Although this paper showed that the EOs of *O. sanctum*, *O. basilicum*, *O. gratissimum*, *M. piperita*, *M. arvensis*, *T. erecta*, and *L. angustifolia* all had different antifeedant effects on *A. gossypii*, in-depth studies are needed to determine their individual antifeedant mechanisms. We will identify the active ingredients in plant essential oils in subsequent research and further evaluate which components in the plant essential oils have antifeedant effects on cotton aphids. *A. gossypii*, as a major pest of cotton, has developed a high level of resistance to conventional chemical control, and botanical antifeedants are a novel means for its control. Antifeedants appear to both reduce colonization of plants by reducing their attraction for pests and reduce feeding of any pest individuals that do begin to attack the host plant, slowing their population growth. EOs protect the host plant and also avoid harming natural enemies and other non-target insects [48]. EOs can, therefore, be used in combination with biological control, natural enemy conservation, and other such techniques to better promote sustainable management of *A. gossypii* in cotton fields.

Currently, botanical insect antifeedants are not yet used on a large scale due to costs in their mass production [49]. Plant EOs are also limited by their relatively high volatility, which limits the duration of their efficacy. Technologies that promote the slow, controlled release of EOs by means of microencapsulation [50] using polymer films have potential to increase the persistence of EOs, allowing for and less frequent applications.

## 4. Materials and Methods

### 4.1. Sources of Essential Oils and Test Insects

Essential oils were purchased from Poli Aroma Pharmaceutical Technology Co, Shanghai and the EOs were diluted with acetone to 1 μg/μL for testing. The seven essential oils were extracted from plants using the distillation method. The entire plants of *O. sanctum*, *O. basilicum,* and *O. gratissimum* were subjected to extraction. For *M. piperita* and *M. arvensis*, the extraction was performed on the flowering tops and leaves. For *T. erecta* and *L. angustifolia*, the flowers were the parts used for extraction.

We initiated a colony of *A. gossypii* by collecting wingless aphids from experimental cotton fields (CAAS, 39.53° N, 116.70° E) of the Lang Fang Experiment Station of the Chinese Academy of Agricultural Sciences (Hebei Province, China). The aphid colony was reared in 12-well plates, with each well two-thirds filled with 2% agar on which a cotton leaf disk excised with a leaf punch from uninfested leaves of cotton plants grown in the greenhouse. The colony was held at 26 ± 1 °C, 50 ± 5% rh, a 16 L:8 D photoperiod.

### 4.2. Growth, Development, and Fecundity

Based on Chen’s experimental method [51], fresh round cotton leaves were cut using a hole punch (23 mm diameter), and were then placed in an inverted orientation on each cell of a 12-well cell culture plate on a bed of 2% agar to which we applied 50 µL of plant essential oil at a concentration of 1 µg/µL. According to Liu and Wang ‘s experimental method, we used 12-well cell culture plates for the experiment [52,53]. Following Izakmehri and Castilhos [54,55], we used acetone as the solvent for dissolving the essential oils. The acetone was allowed to evaporate naturally. For observation of nymphal and adult development, adult aphids were transferred onto the leaves. One adult aphid was placed in each cell. After twenty-four hours, one newborn nymph was retained, and the adults and other nymphs removed with a brush. Plates were covered with nylon gauze to prevent aphid escape, but still allow the EO to evaporate naturally. The leaves were replaced every 2 days and the essential oil renewed at the same level. Observations were made once a day to record the molting time of nymphs, the number of nymphs surviving, the daily production of adult aphids and the number of adult aphids surviving until all adult aphids were dead. Each treatment had 50 replicates, 1 insect being a replicate. The experiments were conducted in a constant temperature and light incubator at 26 ± 1 °C, 50 ± 5% rh, and a 16 L:8 D photoperiod.

### 4.3. Effect of EOs on Aphid Feeding Behaviors

To assess the effect of EOs on aphid feeding behavior, we used a Giga-8 DC-EPG device (EPG-systems Wageningen, Netherlands) with a signal converter card (Di 710, EPG-systems, Wageningen, Netherlands), and the Giga-8 DC-EPG was placed in a Faraday cage (80 cm × 60 cm × 60 cm). Vigorous *A. gossypii* adults were selected and starved for 1 h before testing. A wooden stick was inserted vertically into each cotton seedling nutrient bowl (specifications: upper diameter of 12 cm, lower diameter of 9 cm, height of 10 cm). A piece of cardboard larger than the leaf was mounted on the stick, with the height in line with the cotton leaf. The cotton leaf was turned and affixed to the cardboard. The cotton leaf was coated with 1 μg/μL of an EO with an artist’s brush, while the control leaf was coated with acetone. The volume of essential oil used was 500 μL on each leaf. Acetone did not damage the plant tissue. Aphids were gently placed on the lower surface of individual cotton leaf (choosing the third true leaf) after being attached to the gold-wire electrode with conductive silver paint. One aphid was placed on each cotton plant for testing. Recordings continued for 8 h for each recording session, and at the end of the recording, the cotton leaf and aphid were replaced with new ones. The experiment was run in a greenhouse at 26 °C, 75% rh, and continuous light. Each EO treatment and control was replicated more than 10 times [56].

Following Liu and Sandanayaka et al. [28,57], we recorded the following aphid feeding behaviors: (1) non-probing (Np waves), (2) probing in non-vascular tissues (C waves), (3) probing inside cells (Pd waves), (4) secretion of saliva into phloem sieve elements (E1 waves), (5) phloem ingestion (E2 waves), (6) ingestion of xylem sap (G waves), and (7) mechanical difficulty in stylet penetration (F waves).

### 4.4. Honeydew Excretion

To measure aphid feeding, we recorded the amount of honeydew produced as a surrogate parameter using the honeydew color spot area method of Zhang et al. [58], in which bromophenol green reacts with the amino acids in the honeydew to produce a visible spot, whose area is determined. To do so, the qualitative filter paper (9 cm dia) was first soaked with 0.1% bromocresol green solution, an average of 100 mL of solution is soaked in 10 qualitative filter papers, and then dried in an oven at 75 °C. The filter paper was then bright orange. Agar (2%) was then poured into plastic cups (6.5 cm dia), and fresh cotton leaves spread onto the agar to preserve freshness. To each leaf, we applied 500 µL of the plant essential oil solution, at a concentration of 1 µg/µL. Five third instar aphids were placed in each cup (after being starved for 1 h before the test). The plastic cups were then inverted over filter paper so that most of honeydew secreted by *A. gossypii* would fall onto the filter paper, which turned peacock blue on contact with honeydew. After 24 h, the filter paper was removed, and the blue area (mm^2^) was measured with a transparent standard calculation paper. The experiment was conducted in a greenhouse at 26 ± 1 °C, 50 ± 5% rh, and under natural light. Treatments and controls were each replicated 18 times.

### 4.5. Statistical Analyses

Aphid development, reproduction and population parameters were analyzed according to the age-stage hermaphroditic life table theory using the TWOSEX-MS Chart program (Version 2021.10.30) (http://140.120.197.173/Ecology/prod02.htm) (accessed on 1 December 2021) [59,60]. Standard errors for nymphal developmental duration, adult aphid longevity, reproduction, and population parameters were calculated using the bootstrap method within the software, with a bootstrap count of 100,000, and differences between treatments were analyzed using the software’s paired bootstrap test (*p* < 0.05).

The EPG test data were analyzed using Stylet+a software (Version 1.30) for each waveform, Excel was used for simple statistics on the time and numbers of each waveform, and then SPSS 21.0 software was used to assess differences between treatments with one-way analysis of variance (ANOVA). Differences between treatments were compared using the LSD multiple comparisons method (*p* < 0.05).

Evaluation of *A. gossypii* antifeedant activity using antifeedant rate:$$\mathrm{Antifeedant}\;\mathrm{rate}=\frac{\mathrm{Honeydew}\;\mathrm{area}\;\mathrm{of}\;\mathrm{control}\;\mathrm{aphids}-\mathrm{Treatment}\;\mathrm{groups}}{\mathrm{Honeydew}\;\mathrm{area}\;\mathrm{of}\;\mathrm{control}\;\mathrm{aphids}}$$

Antifeedant rates were simply statistically analyzed using Excel and then SPSS 21.0 software was used to assess differences between treatments with one-way analysis of variance (ANOVA). Differences between treatments were compared using the LSD multiple comparisons method (*p* < 0.05).

## 5. Conclusions

In conclusion, we have demonstrated that the EOs of all seven test plants (*O. sanctum*, *O. basilicum*, *O. gratissimum*, *M. piperita*, *M. arvensis*, *T. erecta*, and *L. angustifolia*) affected the growth, development, and reproductive ability of *A. gossypii*, inhibited aphid feeding behavior, and reduced the amount of honeydew excreted daily by *A. gossypii*. The EOs of *O. sanctum*, *M. piperit*a, and *T. erecta* had the greatest effect, followed by EOs of *M. arvensis* and *L. angustifolia*, and finally the EOs of *O. basilicum* and *O. gratissimum*. Thus, among the seven EOs studied, the EOs of *O. sanctum*, *M. piperita*, and *T. erecta* have potential to be developed as antifeedants for the non-pesticide chemical control of *A. gossypii*.

## Figures and Tables

**Table 1 plants-13-00916-t001:** Effects of essential oils of seven plants on the growth and development of *Aphis gossypii*.

EOs	Developmental Times (d) of Life Stages				Pre-Adults
	1st	2nd	3rd	4th	
Control	1.18 ± 0.06 cd	1.09 ± 0.04 d	1.09 ± 0.05 ab	1.44 ± 0.07 ab	4.80 ± 0.09 d
*O. sanctum*	1.29 ± 0.08 bc	1.39 ± 0.07 ab	1.12 ± 0.05 ab	1.40 ± 0.07 abc	5.19 ± 0.07 ab
*O. basilicum*	1.49 ± 0.07 a	1.18 ± 0.06 bcd	1.09 ± 0.04 ab	1.14 ± 0.05 d	4.89 ± 0.09 cd
*O. gratissimum*	1.36 ± 0.07 abc	1.20 ± 0.05 bcd	1.10 ± 0.04 ab	1.22 ± 0.06 cd	4.88 ± 0.11 cd
*M. piperita*	1.34 ± 0.07 abc	1.51 ± 0.14 a	1.21 ± 0.06 a	1.38 ± 0.07 abc	5.23 ± 0.06 a
*M. arvensis*	1.46 ± 0.08 ab	1.16 ± 0.05 cd	1.18 ± 0.05 a	1.38 ± 0.07 abc	5.18 ± 0.11 ab
*T. erecta*	1.53 ± 0.08 ab	1.15 ± 0.05 d	1.10 ± 0.04 ab	1.33 ± 0.07 bc	5.10 ± 0.10 abc
*L. angustifolia*	1.06 ± 0.03 d	1.35 ± 0.07 abc	1.00 ± 0.00 b	1.54 ± 0.07 a	4.96 ± 0.05 bcd
	*F*_7, 385_ = 4.760	*F*_7, 375_ = 3.954	*F*_7, 373_ = 1.885	*F*_7, 371_ = 3.332	*F*_7, 371_ = 3.339
	*p <* 0.001	*p <* 0.001	*p* = 0.071	*p* = 0.002	*p* = 0.002

Note: Values are means ± SE. Data in the same column followed by different lowercase letters differed significantly (*p* < 0.05).

**Table 2 plants-13-00916-t002:** Effects of essential oils of seven plants on the longevity and fecundity of *Aphis gossypii*.

EOs	Adult Longevity (d)	Reproductive Days (d)	Fecundity
Control	29.67 ± 0.67 a	14.04 ± 0.32 a	53.18 ± 0.99 a
*O. sanctum*	26.25 ± 1.12 b	14.21 ± 0.78 a	41.69 ± 2.44 cde
*O. basilicum*	26.36 ± 0.89 b	11.30 ± 0.36 b	49.55 ± 1.88 ab
*O. gratissimum*	26.76 ± 0.75 b	11.08 ± 0.34 b	49.24 ± 1.27 ab
*M. piperita*	25.32 ± 1.16 bc	13.47 ± 0.82 a	39.53 ± 2.6 de
*M. arvensis*	25.80 ± 0.98 b	10.81 ± 0.36 b	46.20 ± 1.95 bc
*T. erecta*	22.92 ± 0.85 c	11.25 ± 0.44 b	43.79 ± 1.46 cd
*L. angustifolia*	24.25 ± 0.99 bc	13.55 ± 0.66 a	36.63 ± 2.12 e
	*F*_7, 371_ = 4.27	*F*_7, 367_ = 7.219	*F*_7, 371_ = 8.390
	*p <* 0.001	*p <* 0.001	*p <* 0.001

Note: Values are means ± SE. Data in the same column followed by different lowercase letters differed significantly (*p* < 0.05).

**Table 3 plants-13-00916-t003:** Effects of essential oils of seven plants on the life table parameters of *Aphis gossypii*.

EOs	*R* _0_	*T* (d)	*r* _m_	*λ*
Control	47.86 ± 2.41 ab	8.99 ± 0.16 c	0.4304 ± 0.0099 a	1.5379 ± 0.0152 a
*O. sanctum*	40.02 ± 2.59 cde	10.24 ± 0.15 a	0.3605 ± 0.0072 c	1.4340 ± 0.0104 c
*O. basilicum*	43.60 ± 2.80 abcd	9.27 ± 0.13 bc	0.4071 ± 0.0096 ab	1.5024 ± 0.0143 ab
*O. gratissimum*	48.26 ± 1.57 a	9.23 ± 0.15 bc	0.4199 ± 0.0073 ab	1.5218 ± 0.0111 ab
*M. piperita*	37.16 ± 2.77 de	10.04 ± 0.13 a	0.3601 ± 0.0080 c	1.4335 ± 0.0115 c
*M. arvensis*	46.20 ± 1.92 abc	9.55 ± 0.16 b	0.4014 ± 0.0073 b	1.4940 ± 0.0109 b
*T. erecta*	42.02 ± 1.84 bcd	9.25 ± 0.16 bc	0.4043 ± 0.0078 b	1.4982 ± 0.0117 ab
*L. angustifolia*	35.14 ± 2.26 e	9.51 ± 0.13 b	0.3742 ± 0.0068 c	1.4538 ± 0.0099 c

Note: Values are means ± SE. Data in the same column followed by different lowercase letters differed significantly (*p* < 0.05).

**Table 4 plants-13-00916-t004:** Effects of essential oils of seven plants on non-phloem phase feeding behaviors of *Aphis gossypii*, as total duration of behaviors (min) for C, F, G, Np, and Pd behaviors, as well as total number of behavioral events: C-events and Np-events.

Treatment (*n*)	C	F	G	Np	Pd	C Events	Np Events
Control (10)	205.97 ± 18.369 b	6.97 ± 5.416 b	13.00 ± 7.365 d	0.18 ± 0.175 c	13.44 ± 1.443 ab	186.00 ± 17.578 bc	0.20 ± 0.200 c
*O. sanctum* (11)	240.90 ± 16.574 ab	20.69 ± 10.040 ab	58.22 ± 14.374 bcd	34.89 ± 8.350 a	12.58 ± 1.459 b	186.91 ± 20.866 bc	9.64 ± 2.180 a
*O. basilicum* (11)	237.38 ± 25.908 ab	4.52 ± 3.902 b	73.22 ± 21.420 abc	33.46 ± 13.111 a	14.54 ± 1.823 ab	230.82 ± 27.775 ab	7.64 ± 2.636 ab
*O. gratissimum* (11)	281.85 ± 26.821 a	9.55 ± 7.996 b	55.79 ± 14.204 bcd	24.55 ± 7.615 ab	13.19 ± 1.400 ab	172.82 ± 12.156 c	4.00 ± 1.279 bc
*M. piperita* (10)	288.70 ± 26.326 a	14.21 ± 11.502 ab	39.03 ± 11.134 cd	6.32 ± 4.161 bc	15.35 ± 0.971 ab	217.40 ± 12.199 bc	1.40 ± 0.884 c
*M. arvensis* (12)	247.52 ± 23.507 ab	40.96 ± 14.030 a	100.21 ± 24.490 ab	18.06 ± 5.136 abc	14.62 ± 1.635 ab	210.00 ± 22.016 bc	5.58 ± 1.960 abc
*T. erecta* (12)	240.65 ± 10.570 ab	1.77 ± 1.766 b	112.25 ± 21.620 a	14.67 ± 6.030 abc	13.90 ± 1.071 ab	207.25 ± 12.283 bc	3.50 ± 1.222 bc
*L. angustifolia* (12)	297.43 ± 21.609 a	23.36 ± 10.341 ab	21.26 ± 13.919 d	10.67 ± 6.310 bc	17.08 ± 1.337 a	272.08 ± 19.719 a	3.75 ± 2.937 bc
	*F*_7, 81_ = 2.004	*F*_7, 81_ = 2.093	*F*_7, 81_ = 4.114	*F*_7, 81_ = 2.783	*F*_7, 81_ = 1.018	*F*_7, 81_ = 2.786	*F*_7, 81_ = 2.445
	*p* = 0.064	*p* = 0.054	*p* = 0.001	*p* = 0.012	*p* = 0.425	*p* = 0.012	*p* = 0.025

Note: Values are means ± SE. Data in the same column followed by different lowercase letters differed significantly (*p* < 0.05). C = Probing in non-vascular tissues, F = mechanical difficulty in stylet penetration, G = ingestion of xylem sap, Np = non-probing, Pd = probing inside cells.

**Table 5 plants-13-00916-t005:** Effects of essential oils of seven plants on phloem phase of *Aphis gossypii*, as total duration of behaviors (min) for E1, and E2 behaviors, as well as total number of behavioral events: as E1-events and E2-events.

Treatment (*n*)	Time to First E1 (min)	E1	E1 Events	Time to First E2 (min)	E2	E2 Events	Percentage of E1 + E2	Total Number of Single E1
Control (10)	54.58 ± 7.751 b	77.76 ± 15.591 a	15.30 ± 1.633 a	112.76 ± 42.052 ab	162.69 ± 24.491 a	9.10 ± 1.560 a	0.50 ± 0.039 a	2.60 ± 0.562 b
*O. sanctum* (11)	175.03 ± 46.451 a	27.57 ± 6.986 c	8.54 ± 1.951 bc	126.78 ± 41.123 ab	85.15 ± 21.611 b	3.82 ± 1.381 b	0.23 ± 0.047 b	2.91 ± 1.282 b
*O. basilicum* (11)	102.84 ± 26.238 ab	41.86 ± 8.664 bc	12.91 ± 1.781 ab	182.44 ± 30.427 a	75.02 ± 16.794 b	4.27 ± 1.121 b	0.24 ± 0.040 b	6.27 ± 1.602 a
*O. gratissimum* (11)	159.31 ± 40.401 ab	39.22 ± 9.034 bc	8.73 ± 2.067 bc	197.40 ± 53.812 a	55.76 ± 24.627 b	3.73 ± 1.630 b	0.20 ± 0.067 b	3.45 ± 0.857 ab
*M. piperita* (10)	135.14 ± 49.783 ab	50.62 ± 10.683 abc	10.30 ± 2.271 abc	32.75 ± 12.343 b	65.57 ± 28.194 b	3.30 ± 1.257 b	0.24 ± 0.067 b	4.90 ± 1.159 ab
*M. arvensis* (12)	115.89 ± 31.654 ab	25.10 ± 5.532 c	6.75 ± 1.286 c	161.45 ± 38.100 a	33.53 ± 14.907 b	2.33 ± 0.752 b	0.12 ± 0.032 b	3.83 ± 1.086 ab
*T. erecta* (12)	176.36 ± 39.023 a	64.48 ± 17.068 ab	12.00 ± 2.336 abc	154.60 ± 42.219 a	32.15±11.671 b	4.00 ± 1.219 b	0.20 ± 0.052 b	6.33 ± 1.356 a
*L. angustifolia* (12)	150.97 ± 32.706 ab	31.95 ± 7.718 bc	13.67 ± 2.524 ab	144.79 ± 38.275 ab	79.96 ± 21.324 b	5.67 ± 1.734 ab	0.23 ± 0.053 b	5.00 ± 1.030 ab
	*F*_7, 81_ = 1.240	*F*_7, 81_ = 2.880	*F*_7, 81_ = 2.074	*F*_7, 81_ = 1.560	*F*_7, 81_ = 3.791	*F*_7, 81_ = 2.177	*F*_7, 81_ = 4.392	*F*_7, 81_ = 1.509
	*p* = 0.291	*p* = 0.010	*p* = 0.056	*p* = 0.159	*p* = 0.001	*p* = 0.045	*p* < 0.001	*p* = 0.176

Note: Values are means ± SE. Data in the same column followed by different lowercase letters differed significantly (*p* < 0.05). E1 = watery salivation into sieve elements, E2 = phloem sap ingestion.

**Table 6 plants-13-00916-t006:** Effect of essential oils of seven plants on honeydew excretion by *Aphis gossypii*.

EOs	*A. gossypii*	
	Proportion Honeydew/mm^2^	Antifeedant Rate%
Control	11.53 ± 0.809 a	—
*O. sanctum*	7.61 ± 0.853 b	33.40
*O. basilicum*	4.08 ± 0.833 c	64.61
*O. gratissimum*	5.64 ± 0.870 bc	51.08
*M. piperita*	5.69 ± 0.878 bc	50.65
*M. arvensis*	4.01 ± 0.686 c	65.22
*T. erecta*	4.70 ± 0.533 c	59.24
*L. angustifolia*	6.17 ± 0.821 bc	46.49
	*F*_7, 136_ = 9.669	
	*p* < 0.001	

Note: Values are means ± SE. Data in the same column followed by different lowercase letters differed significantly (*p* < 0.05).

## Data Availability

Data are contained within the article.

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
