# Peer review of "Effects of Seven Plant Essential Oils on the Growth, Development and Feeding Behavior of the Wingless Aphis gossypii Glover"

_plants, 2024, doi:10.3390/plants13070916_

Round 1
Reviewer 1 Report
Comments and Suggestions for Authors
Numerous studies have been conducted around the world to assess the potential of essential oils in plant protection. Therefore, this topic is not considered to be new or innovative, and the results of such studies are available in the literature. However, it's surprising that the authors did not investigate the impact of specific compounds contained in these oils on the bionomics of Aphis gossypii. In the introduction, the authors primarily focused on research on compounds rather than essential oils. It would be helpful to know why they chose to do so.
I also have some questions about the methodology used in the study, such as:
1. On what basis were the plants selected for the experiment?
2. Why were studies on the growth, development, and fertility of Aphis gossypii conducted on leaf discs instead of young plants/seedlings? Changing the discs every two days could have affected the test results.
3. If the research was conducted under constant conditions/photoperiod, how were they maintained (greenhouse/phytotron chamber)? Please provide more information about the methodology.
4. EPG observations were only carried out in 10 repetitions. This is a very small number of repetitions for this type of research.
5. The methodology used to obtain honeydew secreted by Aphis gossypii after feeding on plants treated with essential oils does not, in my opinion, fully demonstrate their antioxidant properties. Other tests showing their detergent properties could also be used. It would be helpful to explain why such tests were used and whether they should be supplemented.
The summary presented by the authors is an overinterpretation of the obtained results. If the authors develop recommendations for essential oils, the results of all experiments should be considered.
Author Response
Numerous studies have been conducted around the world to assess the potential of essential oils in plant protection. Therefore, this topic is not considered to be new or innovative, and the results of such studies are available in the literature. However, it's surprising that the authors did not investigate the impact of specific compounds contained in these oils on the bionomics of Aphis gossypii. In the introduction, the authors primarily focused on research on compounds rather than essential oils. It would be helpful to know why they chose to do so.
Answer: Thank you for your suggestion. The damage caused by cotton aphids is extremely serious. Due to the overuse of chemical pesticides, the resistance of cotton aphids to pesticides is becoming increasingly severe. Therefore, we urgently need to find effective and non-pesticidal control measures. Using repellents to prevent pest infestation is an extremely important strategy. This method does not cause damage to the host plant and is non-toxic to natural enemies. However, research on cotton aphid antifeedant has not yet received widespread attention and systematic research.
At present, the research on the potential value of plant essential oils for aphid control is quite extensive, and many studies have noted the fumigation effects and toxicity of plant essential oils on aphids. However, few studies have examined the antifeedant activity of plant essential oils on aphids. Therefore, we screened essential oils that have known antifeedant effects on cotton aphids, on the assumption that this study will provide a better theoretical basis for the subsequent development of more chemically defined aphid antifeedant.
We plan to identify the active ingredients in the plant essential oils that we tested in subsequent research and further evaluate which components in these plant essential oils have repellent effects on cotton aphids. We have also expanded our comments on the research progress on the use of plant essential oils in pest control in the introduction. Lines 80-91
I also have some questions about the methodology used in the study, such as:
- On what basis were the plants selected for the experiment?
Answer: Thanks. Plants from the Lamiaceae and Asteraceae families have significant potential for use pest control, and previous research indicated that the essential oils from some species in these families exhibit antifeedant effects on various species of aphids, including Aphis. spp., Lipaphis erysimi Kaltenbach, and Rhopalosiphum maidis Fitch. Consequently, we sought to determine whether the essential oils from Lamiaceae and Asteraceae plants exhibit antifeedant effects on cotton aphids. We conducted preliminary experiments with several species of Lamiaceae and Asteraceae plants whose EOs are commercially available. Based on those preliminary tests, we selected seven essential oils for this study. We have added some comments on this in the Introduction. Lines 92-102
- Why were studies on the growth, development, and fertility of Aphis gossypii conducted on leaf discs instead of young plants/seedlings? Changing the discs every two days could have affected the test results.
Answer: Thanks. The leaf disc method has been widely used in the study of aphid growth and development. We conducted preliminary experiments that confirmed the feasibility of using the leaf disc method in our research, and we have added references to their use in the revised text. Lines 305-306
- If the research was conducted under constant conditions/photoperiod, how were they maintained (greenhouse/phytotron chamber)? Please provide more information about the methodology.
Answer: Thanks for your suggestion. In our research on the growth, development, feeding behavior, and honeydew secretion of cotton aphids, we have consistently set the experimental environment to match the conditions used to rear cotton aphids. For the studies on growth and development, we used a constant temperature and light incubator. For the research on feeding behavior and honeydew secretion, we employed a constant temperature in a greenhouse. All conditions were maintained at 26 ± 1℃, 50 ± 5% rh, and a 16:8 (L:D) photoperiod. We have added some clarification in the Methods on this point. Lines 316-318, 334-335, 356-357
- EPG observations were only carried out in 10 repetitions. This is a very small number of repetitions for this type of research.
Answer: Thanks for your suggestion. In the literature, on the study of aphid feeding behaviors, many other studies also used the same number of replications, which have shown good consistency. Therefore, we believe that the level of replication that we use was reasonable. We have added references on this point in the revised text. Lines 336
- The methodology used to obtain honeydew secreted by Aphis gossypii after feeding on plants treated with essential oils does not, in my opinion, fully demonstrate their antioxidant properties. Other tests showing their detergent properties could also be used. It would be helpful to explain why such tests were used and whether they should be supplemented.
Answer: Thanks for your suggestion. The secretion of honeydew by aphids is closely linked to their feeding behavior and population dynamics, serving as a crucial indicator in the study of aphid feeding behavior. Insects have evolved a comprehensive osmoregulatory mechanism to control their internal osmotic pressure when feeding on the phloem sap of host plants, expelling some of the ingested sugars as honeydew. By quantifying the volume of honeydew excreted by cotton aphids, we were able to determine if plant essential oils influenced the level of cotton aphid feeding. The honeydew secreted by aphids is colorless and transparent, making it challenging to identify with the naked eye. However, many studies have indicated that the honeydew spot method can effectively monitor aphid honeydew secretion. Given that the primary constituents of honeydew are sugars and amino acids, they can react with bromophenol green to produce color spots. Consequently, in our study we estimated the volume of honeydew produced by aphids by measuring the area of these color spots. We have added a discussion of this point in the revised text. Lines 240-242,344-346
The summary presented by the authors is an overinterpretation of the obtained results. If the authors develop recommendations for essential oils, the results of all experiments should be considered.
Answer: Thanks for your suggestion. This study investigates the impact of seven types of plant essential oils on the growth, development, and feeding behavior of cotton aphids, utilizing life tables and insect feeding behavior research techniques (EPG). Additionally, we measured the honeydew secretion of cotton aphids to further validate the antifeedant activity of plant essential oils. We found that the seven types of plant essential oils tested showed varying antifeedant effects on cotton aphids. Upon comprehensive analysis, the essential oils of Ocimum sanctum L., Mentha piperita L. and Tagetes erecta L. were most effective, followed by the essential oils of Mentha arvensis L. and Lavandula angustifolia Mill., and finally the EOs of Ocimum basilicum L. and Ocimum gratissimum L.
Reviewer 2 Report
Comments and Suggestions for Authors
Authors have studied the effects of essential oils (EOs) from seven plant species on the life parameters of a serious insect pest, the wingless cotton aphid, Aphis gossipii. Such kind of research work becomes more and more important, since most pest insects have developed resistances against commercial chemical insecticides.
The present experiments were carefully planned and the results indicate that most of the EOs studied promise a successful use in ecologically friendly pest control.
A serious problem, however, is that authors have used commercially available EOs without knowing (or presenting) the chemical composition of the oils. It is well known, the the chemical composition of EOs from one and the same plant may differ even more than between different plant species, due to different plant parts used for extraction, to extraction methods used or to different season, plants were harvested. This makes the results at least questionable. It is imperative to show the chemical compositions of the EOs used here.
Another inadequacy of the manuscript is that authors do not say, which amount of essential oils were applied onto the leaves. The same applies to the number of aphids transferred onto the leaves.
Minors:
- line 37: area covered
- line 81: alkyl-substituted
- line 84: ß-Thujone in uppercase letters
- line 88: can possibly be used
- line 91: and not in italics
- line 92: delete one "analyzed"
- line 99: were affected
- line 104: have effects
- line 124: species name in italics
- Tables 4 and 5: all species names are not correctly written; what means 2 C and 2 E in the figure legends?
- line 182: can affect
- line 183 the EO of/from
- line 190: delete one !significant increase"
- line 227: delete one "it"
- line 276: delete one "collected"
- line 287: references are not correct
- line 306: 1ug/uL is a concentration, but not an amount
- lines 307 - 316: the method described here has been presented in many former papers. Give a reference and shorten the description
- line 327: delete "by"
- line 448: give the correct species name
- line 464: give the species names correctly
- line 491: insert a space before the bracket
Comments on the Quality of English Language
English needs only minor corrections.
Author Response
Authors have studied the effects of essential oils (EOs) from seven plant species on the life parameters of a serious insect pest, the wingless cotton aphid, Aphis gossypii. Such kind of research work becomes more and more important, since most pest insects have developed resistances against commercial chemical insecticides.
The present experiments were carefully planned and the results indicate that most of the EOs studied promise a successful use in ecologically friendly pest control.
A serious problem, however, is that authors have used commercially available EOs without knowing (or presenting) the chemical composition of the oils. It is well known, the the chemical composition of EOs from one and the same plant may differ even more than between different plant species, due to different plant parts used for extraction, to extraction methods used or to different season, plants were harvested. This makes the results at least questionable. It is imperative to show the chemical compositions of the EOs used here.
Answer: Thanks for your suggestion. The seven essential oils were procured from Poly Fragrance Medical Technology (Shanghai) Co., and all were extracted from plants using the distillation method. The entire plants of Ocimum sanctum L., Ocimum basilicum L., Ocimum gratissimum L. were subjected to extraction. For Mentha piperita L., and Mentha arvensis L., the extraction was performed on the flowering tops and leaves. For Tagetes erecta L. and Lavandula angustifolia Mill., the flowers were the parts used for extraction. We have added some discussion of this point in the revised text. Lines 290-294
Another inadequacy of the manuscript is that authors do not say, which amount of essential oils were applied onto the leaves. The same applies to the number of aphids transferred onto the leaves.
Answer: Thanks. In the growth and development experiment, we utilized a 12-well plate, applying 50µl of plant essential oil at a concentration of 1 µg/µl to each well, and introduced one aphid into each well. For the EPG experiment, we conducted it on the entire leaf of a 4-leaf stage cotton plant of a certain age, applying 500 µl of plant essential oil at a concentration of 1 µg/µl, and attached one aphid to the cotton leaf. The honeydew experiment, we used the entire cotton leaf in a petri dish, applying 500 µl of plant essential oil at a concentration of 1 µg/µl, and introduced five aphids into each dish. We have added this information in the revised text. Lines 304-305, 309, 330,350-351
Minors:
- line 37: area covered
Answer: Thanks and accepted. Line 38
- line 81: alkyl-substituted
Answer: Thanks, and we have deleted this sentence.
- line 84: ß-Thujone in uppercase letters
Answer: Thanks, and we have deleted this sentence.
- line 88: can possibly be used
Answer: Thanks, and we have deleted this sentence.
- line 91: and not in italics
Answer: Thanks and accepted. Line 105
- line 92: delete one "analyzed"
Answer: Thanks and accepted. Line 106
- line 99: were affected
Answer: Thanks and accepted. Line 113
- line 104: have effects
Answer: Thanks and accepted. We changed the tense. Line 118
- line 124: species name in italics
Answer: Thanks and accepted. Line 138
- Tables 4 and 5: all species names are not correctly written; what means 2 C and 2 E in the figure legends?
Answer: Thanks. We have now corrected the species names in Tables 4 and 5. The number "2" is the superscript. We have corrected that mistake. Line 160, 175
- line 182: can affect
Answer: Thanks and accepted. Line 194
- line 183 the EO of/from
Answer: Thanks and accepted. Line 195
- line 190: delete one !significant increase"
Answer: Thanks and accepted. Line 202
- line 227: delete one "it"
Answer: Thanks and accepted. Line 239
- line 276: delete one "collected"
Answer: Thanks and accepted. Line 295
- line 287: references are not correct
Answer: Thanks. These two papers used acetone as a solvent in their tests, and that use had no effect on the results. So we also chose acetone as the solvent for dissolving the essential oils. We have reworked this statement in the Methods. Lines 306-307
- line 306: 1ug/uL is a concentration, but not an amount
Answer: Thanks, all experiments conducted in this study utilized essential oils at a concentration of 1 µg/µl. In the growth and development experiment, 50 µl was applied to each well. For the EPG and honeydew experiments, 500 µl were applied to each leaf. We have added this information in the revised text. Lines 304, 329, 351
- lines 307 - 316: the method described here has been presented in many former papers. Give a reference and shorten the description
Answer: Thanks a lot, and we have shortened the writing as suggested. Lines 330-332
- line 327: delete "by"
Answer: Thanks, and accepted. Line 344
- line 448: give the correct species name
Answer: Thanks, and accepted. Line 477
- line 464: give the species names correctly
Answer: Thanks, and accepted. Line 492
- line 491: insert a space before the bracket
Answer: Thanks, and accepted. Line 527
Comments on the Quality of English Language
English needs only minor corrections.
Answer: Thanks and accepted.
Round 2
Reviewer 2 Report
Comments and Suggestions for Authors
Thank you for revision. A few more typing errors (e.g., in References) can be corrected during proof-reading.